# *Aeromonas trota* Is Highly Refractory to Acquire Exogenous Genetic Material

**DOI:** 10.3390/microorganisms12061091

**Published:** 2024-05-28

**Authors:** Jorge Erick Otero-Olarra, Gilda Díaz-Cárdenas, Ma Guadalupe Aguilera-Arreola, Everardo Curiel-Quesada, Abigail Pérez-Valdespino

**Affiliations:** 1Department of Biochemistry, Escuela Nacional de Ciencias Biológicas del Instituto Politécnico Nacional, Prolongación de Carpio y Plan de Ayala S/N, Col. Santo Tomás, Mexico City 11340, Mexico; olarra.erick@gmail.com (J.E.O.-O.); gilda.diaz.siguelive@gmail.com (G.D.-C.); 2Department of Microbiology, Escuela Nacional de Ciencias Biológicas del Instituto Politécnico Nacional, Prolongación de Carpio y Plan de Ayala S/N, Col. Santo Tomás, Mexico City 11340, Mexico; lupita_aguilera@hotmail.com

**Keywords:** *Aeromonas trota*, HGT barriers, conjugal transfer, transformation, vesiduction

## Abstract

*Aeromonas trota* is sensitive to most antibiotics and the sole species of this genus susceptible to ampicillin. This susceptibility profile could be related to its inability to acquire exogenous DNA. In this study, *A. trota* isolates were analyzed to establish their capacity to incorporate foreign DNA. Fourteen strains were identified as *A. trota* by multilocus phylogenetic analysis (MLPA). Minimal inhibitory concentrations of antibiotics (MIC) were assessed, confirming the susceptibility to most antibiotics tested. To explore their capacity to be transformed, *A. trota* strains were used as recipients in different horizontal transfer assays. Results showed that around fifty percent of *A. trota* strains were able to incorporate pBAMD1-2 and pBBR1MCS-3 plasmids after conjugal transfer. In all instances, conjugation frequencies were very low. Interestingly, several isoforms of plasmid pBBR1MCS-3 were observed in transconjugants. Strains could not receive pAr-32, a native plasmid from *A. salmonicida*. *A. trota* strains were unable to receive DNA by means of electroporation, natural transformation or vesiduction. These results confirm that *A. trota* species are extremely refractory to horizontal gene transfer, which could be associated to plasmid instability resulting from oligomerization or to the presence of defense systems against exogenous genetic material in their genomes. To explain the poor results of horizontal gene transfer (HGT), selected genomes were sequenced and analyzed, revealing the presence of defense systems, which could prevent the stable incorporation of exogenous DNA in *A. trota*.

## 1. Introduction

The genus *Aeromonas* comprises ubiquitous aquatic organisms capable of inhabiting various types of water, including hostile environments such as contaminated or chlorinated water. These bacteria are able to colonize leeches, insects, fish, mollusks and mammals, including humans [1,2]. Thirty-six species [3] and one additional candidate, *Aeromonas* genomospecies *paramedia* [4,5], have been described. Nineteen species are considered pathogens, but *A. caviae*, *A. dhakensis*, *A. veronii* and *A. hydrophila* are the primary pathogens in 94.5% of clinical isolates. *A. trota* has also been reported, but less frequently [6].

*A. trota* was described in 1991 by Martin Carnahan. Phylogenetic studies based on 16S rRNA gene sequence analysis showed that *A. enteropelogenes* is synonymous with *A. trota* [7,8]. The *A. trota* species description was based on strains isolated from fecal specimens from southern Asia and the USA. The term “trota” means “vulnerable”, this name was chosen because of its susceptibility to ampicillin and carbenicillin [7]. This characteristic highlights how the use of ampicillin-containing media for screening fecal specimens for *Aeromonas* spp. results in a negative selection of *A. trota*, leading to an underestimation of this species [9].

The medical significance of *A. trota* was confirmed with the clinical case of a 28-year-old laboratory worker who accidentally ingested approximately 10^9^ organisms of the reference strain ATCC 49657. He developed self-limiting secretory diarrhea that disappeared after two days without complications [7]. A few years later, a heat-labile and pH-stable diarrhoeagenic toxin produced by *A. trota* strains isolated from diarrheal stools was described [10]. Other virulence factors described in *A. trota* are aerolysin toxin, type IV pili and an extracellular serine protease [11]. *A. trota* has been successively isolated from acute diarrhea [12,13] and other clinical cases including wound infection, sepsis [14] and meningitis [15].

Bacterial chromosomes are complex and dynamic. They suffer modifications over time through mutations, rearrangements and acquisition of foreign genetic material, which result in high genome plasticity [16,17]. Bacteria can acquire genetic material through horizontal gene transfer (HGT), which occurs between cells that do not have an ancestor-descendant relationship [18]. Genetic material is not capable of passing through bacterial membranes and cell walls passively, so it requires transport mechanisms such as conjugation, transformation, transduction and vesiduction [19,20]. HGT frequently endows bacteria with the ability to survive in conditions that could be considered hostile to their development [21]. The clinical relevance of HGT lies in its contribution to disseminating antibiotic resistance genes (ARG) and virulence factors through mobile genetic elements (MGE) like transposons, integrons, plasmids and integrative and conjugative elements [22]. However, HGT often faces barriers that limit the transmission process, such as the lack of natural competence, absence of efficient homologous recombination, host specificity towards phages and plasmids, defense systems like the CRISPR-Cas systems, restriction-modification (RM) systems, abortive infection systems and others [16,23]. Defense systems typically cluster in genomic regions called “defense islands” [24,25,26,27].

*Aeromonas* genome plasticity and dynamics are linked to its adaptation to environmental challenges. Some species host different MGE [28]. *A. salmonicida* is the species that shows the highest occurrence of mobile genetic elements like plasmids, transposons and the ARG encoded by them [29,30]. The fact that *A. trota* is the only species susceptible to antibiotics led us to think that this species could be particularly unable to incorporate exogenous DNA. Therefore, this work focuses on testing HGT on *A. trota* using conjugation, electroporation, natural transformation and vesiduction, as well as analyzing the possible genome-encoded barriers that could impede the HGT process.

## 2. Materials and Methods 

### 2.1. Location and Sample Collection

Samples were collected from seafood (oysters and shrimps) served in street kiosks at Villahermosa, Tabasco, a state located in southeast Mexico. Food samples were collected through the four seasons during a three-year period (2013–2015). All samples were transported in iceboxes to the laboratory and processed the same day.

### 2.2. Isolation and Genus Confirmation of Strains 

Seafood samples (25 g) were homogenized in 225 mL alkaline peptone water and incubated at 37 °C for 24 h. After incubation, a loopful of each culture was streaked onto selective thiosulfate-citrate-bile salts-sucrose (TCBS) plates (Becton Dickinson, Cuautitlán, Mexico). Selected colonies were preserved in cryotubes containing Luria broth (LB) (Dibico, Cuautitlan, Mexico) supplemented with 20% glycerol and stored at −70 °C. Genus identification of isolates was established by basic biochemical tests and MALDI-TOF MS mass spectrometry through VITEK MS IVD v3.0.0 database (bioMérieux, Marcy, l’Etoile, France) [31,32]. All isolates different from *Aeromonas* were discarded.

### 2.3. Multilocus Phylogenetic Analysis (MLPA)

*Aeromonas* species were assigned by MLPA, as reported previously [33]. The PCR amplification products of the genes *gyr*A, *gyr*B and *rpo*D were sequenced by the Sanger method using the ABI3730XL system (Applied Biosystems, Foster City, CA, USA) (Appendix A). Sequences from chromatogram files were edited manually using FinchTV v1.5.0 (The Geospiza Research Team, 2004–2006). The resulting dataset was assembled along with the 37 available GenBank sequences (Appendix A) of the same *Aeromonas* genes, with *Oceanimonas* sp. GK1. as an out-group, using Seaview v5.0 [34]. Gene sequences were concatenated for the phylogenetic analysis, which was performed using the maximum likelihood (ML) method, as implemented in the program PHYML v3.0 [35]. The evolutionary model of nucleotide substitution was obtained using the SMS software tool (http://www.atgc-montpellier.fr/phyml-sms/, accessed on 18 April 2024) [36]. 

### 2.4. ERIC-PCR Analysis

The clonality of *A. trota* isolates was assessed using enterobacterial repetitive intergenic consensus (ERIC-PCR) analysis. Primers are listed in Appendix A [37]. PCR products were separated using 10% polyacrylamide gel electrophoresis, and the resulting band patterns were used to create an absence–presence matrix. Similarity among clones was estimated using the Dice similarity index. The dendrogram was constructed using the unweighted pair group method with arithmetic mean (UPGMA). To evaluate the resulting tree, the cophenetic correlation coefficient was calculated with the program Past v3.0 [38]. 

### 2.5. Antibiotic Susceptibility 

The minimum inhibitory concentrations (MIC) for chloramphenicol, tetracycline, streptomycin, nalidixic acid, trimethoprim, dicloxacillin (Sigma, Naucalpan, State of Mexico, Mexico) and kanamycin (Roche Diagnostics GmbH, Mannheim, Germany) were established by the broth microdilution method using 96-well microtiter plates, following the protocol of the Clinical and Laboratory Standards Institute (CLSI, 2015) [39]. *Escherichia coli* ATCC 25922, *Pseudomonas aeruginosa* ATCC 27853 and *A. hydrophila* 6479 were used as controls [40].

### 2.6. Plasmid Transfer by Conjugation

*E. coli* S17-1 *λpir* RP4, *thi*^−^, *pro*^−^ strain was used as donor of plasmid pBAMD1-2 (4.7 Kb) [41], a suicide plasmid that carries a kanamycin resistance transposable gene, pBBR1MCS-3 (5.2 Kb) [42], a wide-host autonomous plasmid encoding tetracycline resistance and pAr-32 (45 Kb), a native plasmid from *Aeromonas* encoding chloramphenicol resistance [43]. Different *A. trota* strains were used as recipients. Donor and recipient strains were grown in LB at 37 °C with shaking to the stationary phase. Cells were mixed in a 1:2 ratio (100 µL of donor and 200 µL of recipient) and the conjugation mixture was placed onto a sterile nitrocellulose membrane on a Luria plate. The matting mix was incubated for 4 h at 37 °C [44]. Cells were recovered and washed to eliminate the nutrients. Transconjugants were selected in M9 minimal medium plates (to select against donor cells) supplemented with 25 μg/mL of kanamycin, 15 μg/mL of tetracycline or 25 μg/mL of chloramphenicol after 48 to 72 h incubation. To confirm transposition after conjugal transfer of pBAMD1-2, genomic DNA of putative transposants was used for PCR amplification of the *aph*A gene (Appendix A). Plasmid DNA from pBBR1MCS-3 transconjugants was extracted using FavorPrep Plasmid DNA Extraction Mini Kit (Favorgen Biotech Corp., Ping Tung, Taiwan). *A. caviae* 6548 (strain from our collection) was used as a positive recipient control for conjugation. Conjugation frequency was calculated by dividing colony former units/mL (CFU/mL) of transconjugant cells by CFU/mL of recipient cells [45].

### 2.7. Transformation by Electroporation

Electroporation of *A. trota* was done following the protocol of Dallaire-Dufresne et al., with some modifications [46]. Competent cells were prepared as follows: a colony of bacteria was inoculated in 1 mL LB. After 18 h, cells were transferred to 100 mL fresh LB and incubated at 37 °C by shaking at 200 rpm until they reached an OD600 of 0.5. Bacteria were washed three times with sterile cold 10% glycerol, concentrated 333-fold and aliquots of 50 μL were kept frozen at −70 °C. Electroporation was performed by mixing 50 μL of competent cells and 500 ng of plasmid DNA in 0.1 cm gap electroporation cuvettes using a Bio-Rad MicroPulser (Hercules, CA, USA) with a voltage of 1.8 kV. Plasmids used were pRANGER BTB-3 (autonomous, non-mobilizable, 3.6 Kb plasmid) [47], pBAMD1-2 and pBBR1MCS-3. After electroporation, bacteria were resuspended in 1 mL LB, incubated at 37 °C for 40 min and plated onto Luria plates supplemented with the appropriate antibiotic. Plates were incubated at 37 °C for 24 to 48 h and transformation efficiency was calculated. *E. coli* DH5α (Invitrogene, Carlsbad, CA, USA) and *A. caviae* 6548 strains were used as positive controls.

### 2.8. Natural Transformation

Natural transformation was performed as described previously in *Aeromonas* spp. [48]. Briefly, bacteria were grown in 20% nutritive broth (NB), pH 7 at 30 °C with shaking for 24 h. Cultures were diluted 1:100 with 20% NB and incubated at 30 °C for another 24 h. Transformation assays were performed in 1.5 mL tubes containing 100 μL of transformation buffer (53.5 mM Tris, pH 8, 20 mM MgSO4, and 50 mM NaCl), 40 μL of cells containing 10^7^ CFU and 1 µg of chromosomal *A. trota* 9.12 transposant DNA with the *aph*A resistance marker. The transformation mixture was incubated at 30 °C for 30 min statically. After incubation, the transformation was terminated by incubating with 3 μL of 1 µg/mL pancreatic DNase at 30 °C for 1 h. The mixture was spread onto Luria agar plates supplemented with 25 μg/mL of kanamycin and incubated at 37 °C for 24 to 48 h. *A. caviae* 6548 was used as positive control. Cells with no added DNA were used as negative controls. 

### 2.9. Vesiduction

Outer Membrane Vesicles (OMV) preparation was performed following the protocol reported for *A. hydrophila* [49]. *A. trota* with the *aph*A chromosomal resistance marker, *A. caviae* 6548 and *E. coli* 10G, both carrying pRANGER-BTB3 plasmid, were used as vesicle donors. The presence of DNA inside DNase-treated OMV was assessed by extracting whole genomic DNA as previously described [44]. PCR was carried out to confirm the presence of the *aph*A gene and pRANGER BTB-3 inside the vesicles (Appendix A). Vesicles electron microscopy was done as follows. OMV suspensions were placed on copper grids coated with formvar and dried using filter paper. Samples were left in contact with the grid for one minute. An amount of 2.5% uranyl acetate was added and left in contact for another minute, then it was removed and the grid was left to dry at room temperature. Vesicles were observed under transmission electron microscopy (Zeiss model Libra 120, Oberkochen, Baden-Württenberg, Germany). Images were digitalized with Gatan Digital Micrograph software v3. 

DNA transfer via OMV was carried out as described by Rumbo et al. [50]. After contact between OMV and receptor cells, *A. trota* transformants were selected on Luria agar plates supplemented with the appropriate antibiotic. *E. coli* JM109 was used as a positive control [51].

### 2.10. Genome Sequencing

Four *A. trota* strains were selected for genome sequencing according to their ability to receive DNA. *A. caviae* 6548, a positive control strain, was also sequenced. In brief, DNA was obtained as described for whole genomic DNA extraction [44]. Concentration and purity were determined by NanoDropTM 2000 (Thermo Fisher Scientific, Waltham, MA, USA) and QubitR 2.0 (Life Technologies, Carlsbad, CA, USA). DNA was sequenced using NEBNext Ultra II Library Prep Kit (NEB, Ipswich, MA, USA). Adapters were ligated to each sample for library construction. Libraries were pooled in equimolar concentrations for multiplexed sequencing on the Illumina NovaSeq platform with 2 × 150 bp run parameters. To check the quality of sequence libraries, FastQC v0.11.8 [52] was used. When necessary, Trimmomatic program v0.39 [53] was used to trim and filter raw paired readings. Reads were de novo assembled using Spades v3.13.0 [54] and quality control of assemblies was tested with QUAST v5.2 [55]. Contigs were organized using Mauve software v2.4.0 [56]. Assembled genomes were annotated using Prokka v1.14.6 [57] and RAST v2.0 [58]. Genome maps were generated with Proksee using the CGView Java script (https://proksee.ca/, accessed on 29 March 2024) [59].

### 2.11. Phylogenomic Analysis

Phylogenomic analysis was performed using the VAMPhyRE software (https://biomedbiotec.encb.ipn.mx/VAMPhyRE/, accessed on 29 March 2024) [60]. The genome distance matrix obtained (Appendix A) was used to construct a phylogenomic tree using the Neighbor-Joining method with MEGA v11 [61]. To establish phylogenomic relationships of *Aeromonas* strains, whole genome comparisons by in silico DNA–DNA hybridization (*is*DDH) and average nucleotide identity (ANI) were carried out. *is*DDH was calculated with the online program GGDC v3.0 [62]. Strains with *is*DDH values ≥ 70% between genomes are considered to belong to the same species. ANI was calculated with ANI calculator. ANI values ≥ 95% between genomes are considered to correspond to the same species [63]. NCBI reference genomes were used to establish sequence comparisons (Appendix A). 

### 2.12. Genome Analysis

Defense systems within *Aeromonas* genomes were predicted with PADLOC web server v2.0.0 [64] and DefenseFinder v1.2.0 [27]. Phage genomes were searched using the PHASTEST v3.0 web server [65]. Plasmids and insertion sequences (IS) were explored using Plasmid Finder v2.1.6-1 [66] and ISEScan v1.7.2.3 [67] programs, respectively. Resistance genes were detected using ARMFinderPlus v3.12.8 with database v2024-01-31.1 [68].

## 3. Results

### 3.1. Phenotypic and Genotypic Identification of A. trota Strains

Twenty-four strains confirmed to belong to the *Aeromonas* genus were subjected to MLPA analysis to establish species. The concatenated *gyr*A, *gyr*B and *rpo*D gene sequences led to a phylogenetic tree reconstruction. Two isolates identified as *A. allosaccharophila* and *A. dhakensis* were discarded for this study, therefore, 22 *A. trota* isolates were considered for further analysis. Evaluation of the clonality of the *A. trota* strains showed that 14 of them corresponded to different clones according to the cophenetic correlation coefficient (0.8035) obtained (Figure 1 and Appendix A). To confirm their identity, strains whose genome was sequenced were subjected to *is*DDH and ANI analysis. *is*DDH values were above 75% in comparison with the reference genomes. ANI values above 97% were observed in all instances. The phylogenomic tree is shown in Appendix A. All *A. trota* isolates, except for 9.7, were unable to grow in Luria plates supplemented with ampicillin (25 µg/mL).

### 3.2. Resistance Profiles

The MIC for antibiotics was determined to confirm that *A. trota* strains have the species-characteristic sensitive profile. As expected, all strains were sensitive to seven antibiotics tested, except for dicloxacillin (Appendix A).

### 3.3. Plasmid Conjugal Transfer

DNA transfer by conjugation in *A. trota* was tested using pBAMD1-2, pBBR1MCS-3 and pAr-32. Fifty percent (*n* = 7) of strains were able to integrate the transposon after mating with the pBAMD1-2 transposon donor. Transposants were confirmed by the presence of *aph*A gene. In mattings with pBBR1MCS-3, thirty-six percent (*n* = 5) of the strains were able to accept the plasmid albeit at low frequency (Table 1). To demonstrate pBBR1MCS-3 conjugal transfer, plasmids from transconjugants were extracted. Gel electrophoresis showed the presence of several plasmid bands. Bands were confirmed as multimers since they yielded a single band after linearization with SmaI (Figure 2). Additional confirmation for the presence of pBBR1MCS-3 in transconjugants was done by PCR amplification of the *tet*C gene. No *Aeromonas* strain was able to accept the pAr-32 plasmid.

### 3.4. Electroporation, Natural Transformation and Vesiduction

Electroporation was tested on *A. trota* strains with 500 ng of plasmids pRANGER BTB-3, pBAMD1-2 and pBBR1MCS-3. All strains were unable to receive any of the three plasmids, with the exception of *A. caviae* 6548, which was transformed with pRANGER-BTB3 and pBBR1MCS-3. *A. trota* strains were also tested for natural transformation by chromosomal DNA. As before, all strains were unable to incorporate DNA in this way. 

The occurrence of DNA in DNase-treated OMV preparations was assessed by agarose gel electrophoresis. The presence of resistance markers was confirmed by PCR amplification (Appendix A). Vesiduction was intended with OMV containing chromosomal DNA or pRANGER BTB-3. All attempts of DNA transfer via vesiduction were unsuccessful.

### 3.5. Identification of Foreign Genetic Elements and Putative Elements That Prevent HGT

Four selected genomes were sequenced and assembled. Sequences were deposited in the NCBI database (Appendix A). Sequences were searched for the presence of defense systems to make comparisons among permissive and refractory strains (Table 2, Figure 3). Interestingly, all *A. trota* strains contain more defense systems than *A. caviae* 6548, which was receptive to DNA transfer. In contrast, *A. trota* strains 5.9, 9.1 and 9.3, which were completely unable to incorporate DNA, showed a wider diversity of defense systems. However, no clear-cut difference between refractory and permissive *A. trota* strains was observed. 

Intuitively, the ability of *A. trota* to receive foreign DNA elements would correlate positively with the presence of plasmids, transposons, and prophages. Genome analysis disclosed the absence of plasmids in all strains. This was confirmed by gel electrophoresis. IS were found in all *Aeromonas* genomes in a limited number. At least three prophages were found within chromosomes of *A. trota* 9.12 and *A. caviae* 6548 strains, whereas phages were absent or scarce in *A trota* 5.9, 9.1 and 9.3. Genomes were analyzed to search for competence genes relating to the *Acinetobacter* model. Nineteen out of twenty-one genes were found, lacking *pil*A and *com*E genes. Finally, the four *A. trota* strains sequenced harbor *bla*_TRU_, which confers cephalotin resistance, while *A. caviae* 6548 harbors *bla*_OXA_ and *bla*_MOX_ genes. These results are summarized in Table 3.

## 4. Discussion

Bacteria belonging to the *Aeromonas* genus are widely distributed and commonly isolated from seafood [6,79,80]. Among the most prevalent *Aeromonas* species isolated from this source are *A. salmonicida*, *A. media*, *A. bestiarum*, *A. veronii*, *A. hydrophila*, *A. dhakensis* and *A. caviae* [81,82,83]. Results in this work showed that *A. trota* was the main species isolated from seafood. The use of ampicillin in selective media to isolate *Aeromonas* species leads to a bias against *A. trota*, since it is known that it is the only species sensitive to this antibiotic, which could explain its low frequency of detection in other works [84,85]. Ampicillin sensitivity also correlates with our findings on the resistance profile found in our *A. trota* collection, where all strains were sensitive to most antibiotics tested. This susceptibility profile and the scarce reports associated with MGE suggest that this species could be particularly incapable of acquiring foreign DNA.

Resistance to antibiotics is one of the main characteristics acquired by HGT [28]. In this study, the most frequent methods to transfer ARG to *A. trota* strains were conducted. In conjugation, 36% of the strains were able to receive pBBR1MCS-3, a broad-host-range plasmid that can be maintained autonomously [42]. Accordingly, *A. trota* transconjugants contained pBBR1MCS-3. However, plasmid profiles revealed the presence of multimers. Plasmid multimerization is driven by homologous recombination and can affect plasmid stability [86,87]. The presence of these multimeric forms of pBBR1MCS-3 could be related to the inability of *A. trota* to maintain the plasmid, since pBBR1MCS-3 lacks a multimer resolution system [88]. Plasmid multimerization decreases copy number, thereby causing instability, as reported in *E. coli* [89]. The second plasmid transferred was pBAMD1-2, a suicide plasmid that can deliver a mini Tn5 transposon carrying the *aph*A gene. When chromosomally integrated, this element is remarkably stable [41]. This characteristic could explain the recovery of more transconjugant-transposed strains (50%) than transconjugants with the autonomous plasmid pBBR1MCS-3 (36%). Conjugation frequencies were in the range of 10^−6^ to 10^−10^ for pBBR1MCS-3 and 10^−5^ to 10^−9^ for pBAMD1-2 in different *A. trota* strains. Some of these conjugation frequencies can be considered very low if compared with different reports in *A. hydrophila*, where conjugation frequencies between 10^−1^ and 10^−6^ have been reported [90,91,92].

*A. trota* strains were subjected to naked DNA transfer either by electroporation or by natural transformation. Electroporation has been studied previously in *A. hydrophila* WQ, where it has been reported to occur at a low transformation efficiency of 4 × 10^2^ CFU/µg DNA [93], whereas in *A. salmonicida* 01-B526, an efficiency of 1 × 10^5^ CFU/µg DNA has been reported [46]. These are very low transformation efficiencies compared with *E. coli*, which attains a transformation efficiency as high as 10^9^ to 10^10^ transformants per microgram of DNA. These results indicate that the *Aeromonas* genus is, in general, difficult to transform by electroporation. *A. trota* was completely refractory to incorporate foreign DNA, yielding no transformants at all. The different behavior towards conjugal transfer and electroporation suggests the presence of different barriers to DNA entrance. 

Only one report on the natural transformation of *Aeromonas* exists [48], this study showed a transformation frequency of 1.95 × 10^−3^ in different *Aeromonas* species, which contrasts with our results, where no *A. trota* isolate could be transformed with homologous chromosomal DNA from *A. trota* carrying the *aph*A gene. These results indicate that *A. trota* does not develop natural competence, or is unable to integrate DNA, critical steps for this HGT mechanism [16]. Since the genes involved in this phenomenon have not been described in *Aeromonas*, in this work, we searched the genomes for genetic determinants engaged in natural transformation based on *A. baylyi*, a Gram-negative bacterium with 21 competence-related genes, known to undergo natural transformation [78]. Results showed that *pil*A and *com*E are the only genes missing in *A. trota-*sequenced genomes. Likely, the absence of *pil*A gene (or its homologue *tap*A in *Aeromonas* spp.) is the reason for the inability of our strains to receive DNA by natural transformation, since it has been reported that *pil*A is strictly required in *A. baylyi* [78,94]. 

Vesiduction is an HGT mechanism that involves membrane vesicles as vehicles for DNA transfer [20,95]. Although this mechanism has been described in several bacterial species like *Pseudomonas aeruginosa*, *A. baumanii*, *E. coli* and *A. veronii* [50,51,96,97], we found that *A. trota* was unable to receive DNA by this mechanism, despite the presence of plasmid and chromosomal resistance genes within OMV. Inability to receive DNA by vesiduction has been previously reported in *P. aeruginosa* PAO1 [98]. Presumably, the *A. trota* membrane cannot fuse with OMV, or the DNA cannot replicate or integrate into the bacterial chromosome. 

To understand the low capacity of transformation, some *A. trota* chromosomes were sequenced and analyzed. *A. trota* strains 5.9, 9.1 and 9.3 unable to receive DNA by any HGT mechanism showed a variety of defense systems, whereas the 9.12 strain and *A. caviae* 6548 that incorporated DNA through at least one of the mechanisms assayed showed fewer defense systems and probably, these were not sufficient to avoid DNA establishment. Protection elements against foreign DNA establishment tend to cluster in genomic regions called “defense islands” [25]. This is also true in the *Aeromonas* genomes analyzed in this study. Among these, RM systems are the most prevalent in prokaryotic genomes, existing in almost 83% of them, and being RM type II more commonly than others [99,100]. Studies report the presence of RM systems in *Aeromonas* [101,102]. *Aeromonas* genomes sequenced in this work exhibited RM systems, RM type I being the most common and present in two loci in one strain, followed by type IV Restriction system. No type II or III RM systems were found. The search revealed the presence of type I CBASS in *A. trota* 9.1 and 9.3. Gabija system was detected in *A. trota* 5.9. Type I CBASS system is composed of an oligonucleotide cyclase that senses the presence of phage DNA and produces a cyclic oligonucleotide signal that activates an effector protein. The predicted activity of this protein is to form membrane pores, leading to cell death and causing abortive infection [74,103]. On the other hand, Gabija consists of the interaction of a nucleotide sensing nuclease GajA with GajB protein, which is predicted to harbor a helicase domain. These proteins form a complex that cleaves phage DNA [73,104]. This response can explain their protective effect against phages, but it remains unclear if this works for plasmids too. Phosphorotioate modification Dnd system is similar to RM systems, where the DndABCDE protein complex acts as a modification module in a sequence-specific manner, which substitutes the non-bridging oxygen with sulfur in the DNA phosphate backbone, and DndFGH, which acts as a cognate restriction module [76]. A recent report suggests that DndFGH could be a defense system independent of DndABCDE activity [105]. Both DndABCDE and DndFGH were found in two refractory *A. trota* strains (9.3 and 9.1). The Dnd system, as an RM system, provides a kind of innate immune response that could impede plasmid entrance in these *A. trota* strains. Other less well-described defense systems acting against phage infections were detected in our genomes. To date, no reports of these exist in the genus *Aeromonas*. More research is needed to discover the specific role of these defense systems.

Other genetic elements whose presence could be the result of HGT (plasmids, IS and phages) were also searched in sequenced genomes. The absence of these elements could likely be related to the presence of a variety of defense systems. Accordingly, plasmids were absent in all strains, and a relatively low number of IS elements and phage genomes were found in the *A. trota* genomes, whereas these elements, with exception of phages, have been reported commonly in *A. salmonicida*, which is a more transformable species [46,106].

## 5. Conclusions

The premise of this work was that the remarkable antibiotic susceptibility profile of *A. trota* could be related to its incapacity to receive foreign DNA. This study revealed that all *A. trota* isolates tested have a low frequency of HGT, which could be due to multiple factors: (i) Plasmid instability resulting from oligomerization, (ii) an incomplete set of competence genes, and (iii) multiple defense systems. These factors could play a role in impeding the acquisition of new genetic traits.

## Figures and Tables

**Figure 1 microorganisms-12-01091-f001:**
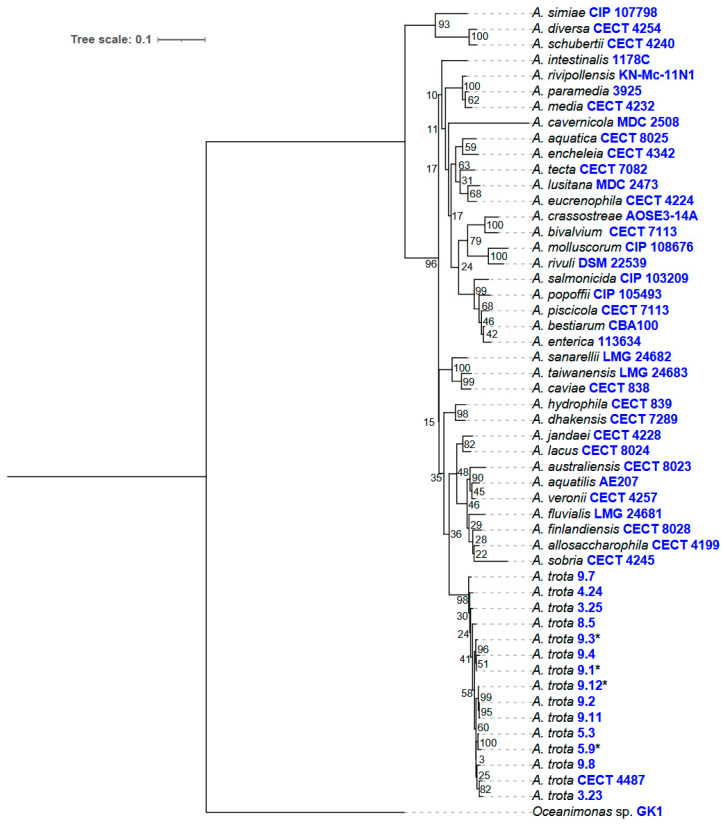
Phylogenetic tree based on Maximum Likelihood method (ML) of the concatenated *gyr*A, *gyr*B y *rpo*D genes. Numbers in the nodes indicate the bootstrap support values (100 replicates), using the Akaike Likelihood Ratio Test (aLRT). The scale bar indicates the number of nucleotidic substitutions per site. * Genomes sequenced in this study.

**Figure 2 microorganisms-12-01091-f002:**
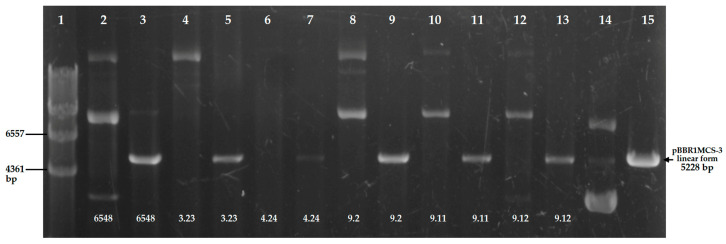
pBBR1MCS-3 plasmid extracted from transconjugant *Aeromonas.* Line 1: λ/HindIII marker. Lanes 2 and 3: supercoiled and SmaI linearized plasmid from *A. caviae* 6548. Lanes 4 to 13 supercoiled and SmaI linearized plasmids from *A. trota* strains, respectively. Lanes 14 and 15 correspond to supercoiled and SmaI linearized pBBR1MCS-3 from *E. coli* S17-1 *λpir*.

**Figure 3 microorganisms-12-01091-f003:**
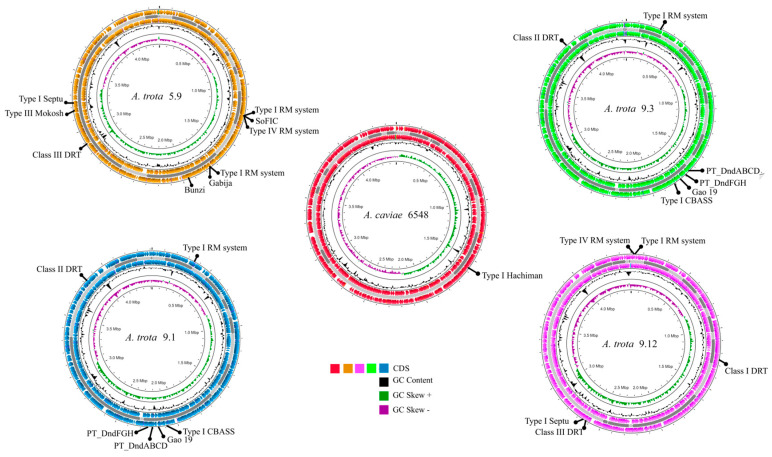
Localization of defense systems in sequenced *Aeromonas* genomes. Each circular map shows contigs organized using Mauve software v2.4.0. The position of the defense systems within genomes is highlighted in the coding sequence circle. Visualization of the genome was done with Proksee.

**Table 1 microorganisms-12-01091-t001:** Conjugal transfer frequencies in *A. trota* strains.

Strain	pBAMD1-2	pBBR1MCS-3
*A. caviae* 6548 *	5.52 × 10^−7^	5.19 × 10^−6^
*A. trota* 3.23	5.44 × 10^−8^	5.94 × 10^−7^
*A. trota* 4.24	2.17 × 10^−6^	3.10 × 10^−6^
*A. trota* 5.3	4.05 × 10^−9^	0
*A. trota* 9.2	2.32 × 10^−7^	6.75 × 10^−10^
*A. trota* 9.8	4.51 × 10^−5^	0
*A. trota* 9.11	4.23 × 10^−7^	8.67 × 10^−7^
*A. trota* 9.12	3.32 × 10^−8^	4.75 × 10^−7^

* Conjugation positive control. *A. trota* 3.25, 5.9, 8.5, 9.1, 9.3, 9.4, 9.7 yielded no transconjugants.

**Table 2 microorganisms-12-01091-t002:** Defense systems detected in *A. trota* genomes.

Strain	Defense Systems	Protein Name
*A. trota* 5.9	Class III Defense-associated reverse transcriptase (DRT) [69]	Drt3a, Drt3b
	Type II Mokosh [70]	MkoC
	Type I Septu [71]	PtuB1, PtuA1
	Type I restriction-modification (RM) system (two loci) [72]	Rease I, MTase I, Specificity I
	Type IV restriction system [72]	mREase IV
	Gabija [73]	GajA, GajB
	Bunzi [70]	BnzA, BnzB
	SoFic [70]	SoFic
*A. trota* 9.1	Class III DRT	Drt1b, RT_UG5-nitrilase
	Type I Cyclic oligonucleotide-based antiphage signaling system (CBASS) [74]	Effector, Cyclase
	Gao 19 [75]	HerA, SIR2
	Phosphorothioate (PT) modification system [76]	DndB-C-D-E and DndF-G-H
	Type I RM system	Rease I, MTase I, Specificity I
*A. trota* 9.3	Class III DRT	Drt1b, RT_UG5-nitrilase
	Type I CBASS	Effector, Cyclase
	Gao 19	HerA, SIR2
	PT modification system	DndB-C-D-E and DndF-G-H
	Type I RM system	Rease I, MTase I, Specificity I
*A. trota* 9.12	Class III DRT	Drt3a, Drt3b
	Type I Septu	PtuB1, PtuA1
	Type I RM system	Rease I, MTase I, Specificity I
	Type IV restriction system	mREase IV
	Class I DRT	Drt4
*A. caviae* 6548 *	Type I Hachiman [77]	HamA1, HamB1

* This strain was permissive to incorporate foreign DNA.

**Table 3 microorganisms-12-01091-t003:** Genetic elements related to HGT in *A. trota* strains sequenced in this work.

Strain	Prophages	Insertion Sequences (IS)		Resistance Genes **	Competence Genes *
		Family	Length (bp)		
*A. trota* 5.9	No	IS3 (1)	1247	*bla* _TRU_	*pil*A and *com*E genes absent.
		IS5 (2)	1062, 1079		
*A. trota* 9.1	Yes (1)	IS5 (2)	1039, 1051	*bla* _TRU_	
		IS256 (1)	1326		
*A. trota* 9.3	No	IS5 (2)	1051, 989	*bla* _TRU_	
		IS256 (1)	1326		
*A. trota* 9.12	Yes (3)	IS5 (3)	1065, 1062, 1490	*bla* _TRU_	
		IS3 (1)	1115		
*A. caviae* 6548	Yes (3)	IS3 (2)	1228, 1318	*bla* _OXA_ *bla* _MOX_	
		IS30 (1)	1974		
		IS481 (1)	1198		

Numbers in parenthesis indicate the number of phage’s genomes or IS. Prophages features are shown in Appendix A. * Compared to *A. baylyi* [78]. ** Accession numbers of the resistance genes: WP_042027926.1, WP_151027527.1, WP_128343957.1.

## Data Availability

The original contributions presented in the study are included in the article/Appendix A, further inquiries can be directed to the corresponding authors.

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
