# Peer review of "Aeromonas trota Is Highly Refractory to Acquire Exogenous Genetic Material"

_microorganisms, 2024, doi:10.3390/microorganisms12061091_

Round 1

Reviewer 1 Report

Comments and Suggestions for Authors

Reviewer’s Comments and Suggestions for Authors

Journal: Microorganisms, MDPI

Manuscript ID: microorganisms-3012996

Type: Article

Title: Aeromonas trota is highly refractory to acquire exogenous genetic material.

Authors: Jorge Erick Otero-Olarra, Gilda Díaz-Cárdenas, Ma. Guadalupe Aguilera-Arreola, Everardo Curiel-Quesada*, and Abigail Pérez-Valdespino* 

The authors of the manuscript Manuscript ID: microorganisms-3012996 focused on testing horizontal gene transfer (HGT) by means of conjugation, electroporation, natural transformation and vesiduction in Aeromonas trota, and analyzing possible genome encoded barriers that could impede the HGT process. A. trota isolates (n = 14) were isolated from seafood and identified, and confirmed the susceptibility to most antibiotics tested. The authors found that half of these A. trota strains were capable to incorporate pBAMD1-2 and pBBR1MCS-3 plasmids after conjugal transfer, with very low conjugation frequencies. A. trota strains showed inability to receive DNA by means of electroporation, natural transformation and vesiduction. Whole genomes of 4 A. trota strains were sequenced and analyzed, which revealed the presence of defense systems which could prevent the stable incorporation of exogenous DNA in A. trota.

Some issues, particularly in the Results, and Materials and Methods sections, should be clarified. Essential extensive revisions of this manuscript should be performed.

Major revisions 

1. The Abstract section is redundancy, please shorten it.

2. The Conclusions section was over-drawn, please rephrase this section, and summarize the major results of this study. For example,

The authors stated that “This study revealed that this species has a low frequency of HGT, that could be due to multiple factors: i) high recombinogenic activity related to plasmid instability”. However, no experimental results in this manuscript provided evidence for this conclusion. Please clarify. 

On the other hand, only limited number of A. trota isolates (n = 14) were analyzed in this manuscript. Therefore, the conclusions were over-drawn for this species.

3. The Results section was preliminary. Certain important data and analyses are missing, for example,

(1) How may raw data were generated from whole genome sequencing of the four A. trota isolates? What were sequencing depths? How many contigs were assembled? ...

(2) What were the general genome features of the four A. trota isolates? For example, genome sizes? G+C content? proteins-encoding genes? ...

(3) The results of all the MICs for antibiotics against the test A. trota isolates should be provided, and presented as a Table.

(4)  No information of the identified prophages in the three genomes of A. trota isolates were provided, e.g., sequence length, predicted genes, ... Please clarify. The authors should further analyze the sequences of the identified prophages. The same issue for the identified ISs.

(5) No results were shown regarding “The occurrence of DNA in DNase treated OMVs preparations was confirmed by agarose gel electrophoresis”. No images of the electron microscopy observation of the prepared OMVs were shown as well. Please clarify.

(6) Lines 278-279: “A. trota strains were also tested for natural transformation by chromosomal DNA. As before, all strains were unable to receive DNA by this way”, please describe the detailed information of the chromosomal DNA.  

4. In the Materials and Methods section, certain essential information was missing. For example,

(7)  The information of the genome annotation was missing, which should be described in the Materials and Methods section.

(8) The detailed information of the used plasmids pRANGER BTB-3, pBAMD1-2 and pBBR1MCS-3 in this study should be added, e.g., the plasmid size.

(9) The method used to determine the “CFU/mL of transconjugants cells“ should be cited.

(10) No information of statistic analysis of the obtained data was described. Please clarify.

5. Please describe the main limitations of this study.

Minor revisions

6. Abbreviations and acronyms are typically defined the first time the term is used within the abstract and again in the main text and then used throughout the remainder of the manuscript. Please consider adhering to this convention, and check throughout the manuscript.

7. Lines 15, 19-20: please rephrase the sentences.

8. Line 23: change to “A. trota strains”. Please check the similar issue throughout the manuscript.

9. Table 2: please provide the references of the defense systems, and provide the gene IDs identified in the A. trota genomes.

10. Figure 3: please describe each of the circles.

11. Table 3: please format this Table, and provide the references of the Resistance genes.

12. Please format the Tables in the Supplementary materials.

13. Please modify as “n = 14”, and check similar problems throughout the manuscript.

14. Lots of English language issues in this manuscript. Please carefully and extensively revise throughout the manuscript.

Comments on the Quality of English Language

Extensive editing of English language required.

Author Response

Dear reviewers, thanks for your observations and comments.

Reviewer 1

  1. Abstract was shortened and modified to avoid redundancies as suggested (Lines 13-29).
  2. Conclusion was modified as suggested (Lines 416-42).
  3. Results
    • Data concerning genome sequencing and assembling were included in Table S5.
    • GC content and CDS were included in Table S5
    • MIC results were incorporated (Table S4).
    • Prophage features were included in Table S6. Prophages and IS are indicators of HGT. We think that in depth characterization is not necessary to this end. We only show results as they are obtained using online phage detection tools.
    • The occurrence of DNA in DNase treated OMV preparations were confirmed by agarose gel electrophoresis. Image is included in Supplementary material. Electron microscopy images of OMV are also included in figure S3.
    • Information on chromosomal DNA used in natural transformation assays was included was incorporated in material and methods (lines 165-166).
  4. Material and methods
    • The information on genome annotation is in line 202.
    • Information on plasmids was included as suggested in lines 126-129 and 153. References are also included.
    • Precisions on the calculation of CFU/mL cells were done (lines 135-136; 142).
    • The conjugal transfer frequency for each recipient was obtained from three independent experiments and no differences between the triplicates were observed.
  5. Discussion includes scope and limitations of the study.
  6. Abbreviations and acronyms were modified as suggested.
  7. Abstract was modified
  8. Line 23. Aeromonas strains were changed to trota strains.
  9. Table 2. References and gene IDs were added.
  10. Caption of figure 3 was modified emphasizing the location of defense systems.
  11. Table 3: References to resistance genes were included.
  12. Tables in the Supplementary materials were modified as suggested
  13. Numbers if strains were modified throughout the manuscript.
  14. English was carefully reviewed

Reviewer 2 Report

Comments and Suggestions for Authors

Otero-Olarra et al examined the potential of 14 Aeromonas trota isolates sampled from seafood to acquire exogenous DNA. MIC testing confirmed the isolates to be susceptible to most of the 8 antibiotics tested with two exceptions (dicloxacillin). Half of the study isolates were able to act as recipients at low frequencies to pBAMD1-2 and pBBR1MCS-3 plasmids by conjugation assays. No strains could accept A. salmonicida pAr-32 plasmid or other exogenous DNA by electroporation, transformation and vesiduction confirming the ability of these strains to avoid horizontal gene transfer. Sequencing of four A. trota genomes detected multiple defense systems suspected to prevent the stable incorporation of foreign DNA in this species.

The study is well designed. In some areas more specific details or references could be provided in the Methods and the Results (original colony isolation i.e. bacterial growth and preliminary identification Aeromonas and/or non-Aeromonas leading to A. trota isolates, origin details, include ERIC PCR gel in Suppl Figure, basis genome selection and whether receptive). This would help the readers identify any association of A. trota strains to a particular food source, confirm clonality (no gel image is provided so difficult), genome content and receptive potential. Including the A. trota type strain for the horizontal transfer assays and in analysis (Table 2, Table 3, Figure 3) would be of benefit for repeating future comparative studies.

Please find the following suggested minor changes and additions to address gaps in the content;

Line 15, 217 establish

Line 22 delete ,

Line 23 rephrase

Line 29 combine sentences

Line 39, 56, 113, 216, 255, 305, 405 delete unnecessary words

Line 44 reference 94.5% end of sentence or [6]

Line 48 The term "trota"

Line 81 A. trota is the only species

Line 88 Food samples

Line 97 Genus identification

Line 112 Uppercase, include type strain and outlier

Line 113 listed in Table S1

Line 115 matrix created by visual analysis?

Line 116, 117 references

Line 129 reference E. coli S17-1 strain, state size of three plasmids

Line 133 recipients, include type strain

Line 137, 254 mating

Line 142 kanamycin resistance gene

Line 145 reference 6548 strain and conjugation potential

Line 173 no added DNA

Line 191 check phrasing

Line 194 ability, high or low?

Line 201 bp

Line 227 Insert details of sampling results from original number of isolates retrieved, suspected Aeromonas isolates, list other Aeromonas species is present, then 14 A. trota. Link species to sample origin, any association with food source or year?

Line 229 species identification

Line 231 past tense contained, 14 isolates

Line 233 Figure S1 include gel image

Line 234 four genomes were sequenced and subjected

Line 236 The 

Line 247 eight antibiotics tested

Line 260 tetracycline resistance determinant

Line 262 include conjugation results (or absence) of selected isolates in Table that were sequenced for easy reference. type strain?

Line 266 state size plasmid, six A. trota transconjugants

Line 269 state plasmid, E. coli donor

Line 281 how did you select which genomes to sequence

Line 292 four genomes and A. caviae control strain, state tool used. Include A. trota type strain? State whether each strain was permissive or refractory in Table 2. Add type strain

Line 295 colour identical label elements to aid matching? include A. trota type strain?

Line 300 did you confirm absence of plasmid DNA by extraction? does strain A. caviae possess a plasmid to use a positive extraction control or include an isolate with known plasmid carriage?

Line 301 three

Line 308 include A. trota type strain

Line 309 typo

Line 310 * in Table?

Line 318 study, no sampling results of other Aeromonas isolates (bacterial identification) were provided

Line 323 eight

Line 324 with

Line 330 the plasmid profiles

Line 346, 348 strain?

Line 355 of; The previous study

Line 360 acting as

Line 374 the

Line 376 , 

Line 377, 378, 402 list strains

Line 382 study

Line 386 with

Line 389 state how many genomes

Line 405 less well described, but it is; they

Line 418 i) a high

Line 419 ii) an incomplete

Line 420 typo

I was unable to upload my notes in the pdf file highlighting edit points and location.

Comments on the Quality of English Language

Minor edits are needed

Author Response

Reviewer 2

Dear reviewers, thanks for your observations and comments.

Minor changes and additions to the manuscript were done as suggested.

Line 224-234. Details of sampling results were included as suggested.

Figures S1 was modified, including gel image.

In this study we did not include an A. trota type strain since no reports on horizontal gene transfer in this strain exist. We analyzed the A. trota CECT 4487 genome to establish the defense systems. This type strain exhibits CBASS, SoFIC, type IV RM and class III DRT. However, we do not have the strain to assay its behavior.

Line 301. Plasmid absence was confirmed by agarose gel electrophoresis.